# MicroRNAs in the Regulation of RIG-I-like Receptor Signaling Pathway: Possible Strategy for Viral Infection and Cancer

**DOI:** 10.3390/biom13091344

**Published:** 2023-09-04

**Authors:** Dengwang Chen, Qinglu Ji, Jing Liu, Feng Cheng, Jishan Zheng, Yunyan Ma, Yuqi He, Jidong Zhang, Tao Song

**Affiliations:** 1Department of Immunology, Zunyi Medical University, Zunyi 563002, China; 18185747211@163.com (D.C.); l1765519987@163.com (J.L.); chengfeng_202308@163.com (F.C.); 18311683502@163.com (J.Z.); 18722715640@163.com (Y.M.); 2School of Pharmacy, Zunyi Medical University, Zunyi 563002, China; jql_carry@163.com (Q.J.); yqhe2016@zmu.edu.cn (Y.H.); 3Collaborative Innovation Center of Tissue Damage Repair and Regeneration Medicine, Zunyi Medical University, Zunyi 563002, China; 4Special Key Laboratory of Gene Detection & Therapy of Guizhou Province, Zunyi Medical University, Zunyi 563002, China

**Keywords:** innate immune, RLR pathway, microRNAs, viral infection, cancer

## Abstract

The retinoic acid-inducible gene I (RIG-I)-like receptors (RLRs) play a crucial role as pattern-recognition receptors within the innate immune system. These receptors, present in various cell and tissue types, serve as essential sensors for viral infections, enhancing the immune system’s capacity to combat infections through the induction of type I interferons (IFN-I) and inflammatory cytokines. RLRs are involved in a variety of physiological and pathological processes, including viral infections, autoimmune disorders, and cancer. An increasing body of research has examined the possibility of RLRs or microRNAs as therapeutic targets for antiviral infections and malignancies, despite the fact that few studies have focused on the regulatory function of microRNAs on RLR signaling. Consequently, our main emphasis in this review is on elucidating the role of microRNAs in modulating the signaling pathways of RLRs in the context of cancer and viral infections. The aim is to establish a robust knowledge base that can serve as a basis for future comprehensive investigations into the interplay between microRNAs and RIG-I, while also facilitating the advancement of therapeutic drug development.

## 1. Introduction

### 1.1. RLR Signaling

#### 1.1.1. The Structural Characterization of RLRs

RLRs are classified within the DExD/H-box RNA helicase family and encompass three distinct members, namely RIG-I, melanoma differentiation-associated gene 5 (MDA5), and laboratory of genetics and physiology 2 (LGP2), which are encoded by the genes *DDX58*, *IFIH1*, and *DHX58*, respectively. These three proteins exhibit 30% similarity in their amino acid sequences, particularly in their key domains (RNA helicase, pincer, and CTD domains) [1]. In general, the central helicase domain and the C-terminal domain of these RLRs are responsible for their RNA recognition capability [2]. The N-terminal region of RIG-I and MDA5 contains a tandem caspase recruitment domain (CARD) (Figure 1). This domain is important for the interaction of these proteins with the adaptor molecule, named the mitochondrial antiviral signaling protein (MAVS; also referred to as VISA, Cardiff, and IPS-I). The MAVS protein possesses a solitary CARD domain at its N-terminal region, which interacts with the N-terminal CARD domain of RIG-I. This interaction prompts the assembly of prion-like aggregates of MAVS proteins, subsequently leading to the activation of the transcription factors nuclear factor kappa B (NF-κB) and interferon regulatory factor 3 (IRF3) [3]. LGP2 is mostly similar to RIG-I and MDA5, with the only difference being that it lacks the N-terminal CARD.

#### 1.1.2. The Recognition of RNAs by RLRs

RIG-I primarily detects two main types of RNA: single-stranded RNAs (ssRNAs) with a triphosphate group (PPP) at their 5′ end, and short double-stranded RNAs (dsRNAs) (<300 bp) [4,5,6,7,8]. Conversely, MDA5 predominantly recognizes long dsRNAs (>1000 bp) [9,10]. Unlike RIG-I and MDA5, LGP2 lacks the CARD structural domain, rendering it unable to recognize RNA. However, studies have demonstrated that LGP2 could be involved in coordinating RIG-I and MDA5 [11,12]. Emerging investigations have shed light on the involvement of LGP2 in the recognition of viral RNA [13,14] and its significant role in self-RNA sensing and RNA silencing [14,15]. Moreover, studies have indicated that RNase L-mediated degradation of parainfluenza viral products preferentially activates MDA5 [4]. Notably, the nucleocapsid proteins of paramyxoviruses effectively shield the viral genomic RNA, safeguarding it from RNase digestion and host proteins like MDA5 [16]. Consequently, the lack of dsRNA detection in paramyxovirus infections by dsRNA recognition antibodies is unsurprising [17]. Furthermore, because the nascent genomic RNA is encapsulated, the dsRNA segment within the replication/transcription complex is limited (>2 kb is regarded as an MDA5 trigger). Additionally, coronaviruses have been found to evade detection by MDA5 through 2′-O-methylation of mRNA [18]. However, it is evident that future advancements should take into account higher-order RNA structure, sequence, and modifications, in addition to length. Notably, recent discoveries regarding the biological function of RIG-I in various cellular activities confirm its ability to interact with different endogenous RNAs (e.g., microRNA [19], panhandle RNA [20], circRNA [21], and snRNA [22]). Given that the majority of viral RNA possesses a 5′-PPP group, the RIG-I receptor primarily recognizes and distinguishes viral infections [23].

#### 1.1.3. The RLR Signal Transduction

Under normal circumstances, RIG-I/MDA5 remains in an inactive and inhibited state within the cytoplasm [24,25]. However, during viral infection, RIG-I/MDA5 recognizes viral RNA, resulting in conformational changes within the RIG-I/MDA5. These conformational changes facilitate the interaction between RIG-I/MDA5 and MAVS on the surface of mitochondria [26]. MAVS dimerization then serves as a scaffold to assemble signaling complexes at the outer mitochondrial membrane, attracting molecules such as tumor necrosis factor receptor-associated factor 6 (TRAF6). Upon recruitment to the receptor-associated interleukin-1 receptor-associated kinase 1 (IRAK1)-IRAK4-MyD88 (myeloid differentiation primary response 88) adaptor protein complex, TRAF6 undergoes lysine 63 (K63)-linked polyubiquitination. This event promotes the recruitment of downstream regulators like transforming growth factor-beta-activated kinase 1 (TAK1) and TAK1-binding protein 2/3 (TAB2/3), which in turn activate NF-κB, leading to the production of inflammatory cytokines [27].

In parallel, linker proteins TRAF family-member-associated NF-κB activator (TANK), nucleosome assembly protein 1 (NAP1), and, similar to NAP1, TBK1 adaptor (SINTBAD), are recruited by the TRAF3 bound to MAVS. TANK binding to TANK-binding kinase 1 (TBK1), which is linked to upstream RLR signaling through TANK, leads to the phosphorylation of IRF3. Phosphorylated IRF3 (p-IRF3) dimerizes and translocates to the nucleus to initiate the transcription of the IFN-β gene [28,29]. This process triggers the expression of antiviral genes; increases the production of pro-inflammatory cytokines and IFN-I [30,31]; and ultimately collaborates with IFNs, inflammatory cytokines, and other antiviral products to exert antiviral effects. Conversely, IFN-I can also provide feedback modulation to the RIG-I signaling pathway. For instance, the IFN-induced long noncoding RNA, lnc-Lsm3b, competes with viral RNA for binding to RIG-I monomers and serves as a feedback mechanism to deactivate RIG-I’s innate activity at a later stage of the innate immune response [32].

#### 1.1.4. Regulation of RLR Signaling by LGP2 and E3 Ubiquitin Ligases

While LGP2 lacks the ability to directly recognize RNA and activate downstream pathways, it plays a crucial role in the regulation of RLR signaling. LGP2 exerts its inhibitory effects on the RIG-I-mediated signaling pathway through various mechanisms. Firstly, the C-terminal RD domain of LGP2 forms a heterodimeric complex with the RD domain of RIG-I, hindering RIG-I multimerization and thus impeding its recognition of viral dsRNAs. Also, LGP2 can hinder the interaction between RIG-I and MAVS, attenuating RIG-I signaling. Furthermore, LGP2 can also dampen RIG-I signaling by inhibiting RIG-I’s binding to MAVS [33]. Moreover, LGP2 demonstrates the ability to inhibit the activation of IRF3, a downstream transcription factor of RIG-I, thereby potentially exerting a negative regulatory effect [34].

Earlier studies have identified several E3 ubiquitin ligases, including Riplet, tripartite motif-containing protein 25 (TRIM25), and ring finger protein 123 (RNF123), which modulate RIG-I activity [35,36]. Initially, TRIM25 was believed to be crucial for the ubiquitination of RIG-I [37]. However, recent investigations have established that Riplet, rather than TRIM25, is responsible for RIG-I ubiquitination. To further comprehend the specific recognition of RIG-I by Riplet and determine the E3 ligase responsible for MDA5 recognition, future studies are warranted [38,39]. In addition to the regulation of RLR signaling through LGP2 and E3 ubiquitin ligases, increasing evidence confirms that microRNAs are also involved in the regulation of RLR signaling [19]. In recent years, there has been a notable surge in research interest dedicated to exploring the regulation of the RIG-I signaling system.

### 1.2. MicroRNAs

The short-length, noncoding RNA molecules referred to as microRNAs (miRNAs) occur in a size range of 19–25 nucleotides and are reported to regulate the post-transcriptional silencing of target genes. A single miRNA can target hundreds of mRNAs simultaneously, thereby affecting the expression of numerous genes that are frequently involved in a functionally interconnected pathway [40]. The majority of miRNAs exist in the form of primary miRNAs (pri-miRNAs), which are transcribed from DNA sequences and typically range in length from approximately 300 to 1000 nucleotides. The processing of pri-miRNAs generates precursor miRNAs (pre-miRNAs), which are approximately 70 to 90 nt in length and are considered the precursor of microRNA. After being cleaved by the Dicer enzyme, the pre-miRNA is re-digested to generate mature miRNA of approximately 19~25 nt in length [41,42,43] (Figure 2).

Two miRNAs named lin-4 and let-7 have been verified in previous studies to be partially complementary to the 3′ region of several messenger RNA (mRNA) targets of interest. The noncoding regions (3′-UTRs) are responsible for the inhibition of protein translation, thereby inhibiting protein synthesis ultimately [44]. MiR-125b, a human counterpart of lin-4, along with let-7, are closely linked to cancer development and hold promise as potential diagnostic and prognostic markers [45,46]. Additionally, microRNAs have been found to play significant roles in the pathogenesis of cardiovascular diseases (CVDs) [47,48], cancer initiation and progression [49,50,51], diabetes [52,53], Alzheimer’s disease [54,55], and inflammation [56,57]. It is worth noting that cellular miRNAs can serve as biomarkers for the diagnosis and monitoring of COVID-19 [58,59,60,61]. Therefore, microRNAs are considered novel means of diagnosis, therapy, and prognosis in the context of the above-stated diseases.

## 2. Regulation of RLR Signaling by microRNAs

### 2.1. The Regulation of RLR Signaling by microRNAs during Viral Infection

#### 2.1.1. MicroRNAs Targeting RIG-I/MDA5 during Viral Infection

RIG-I and MDA5, which are pivotal upstream molecules of the RLR signaling pathway, play crucial roles in response to viral infections and relay signals downstream. Activation of the RLR signaling pathway triggers the production of IFN-I. The antiviral effects of IFNs mainly involve the following major pathways: I. Phosphorylation of IFN-induced proteins and RNA degradation facilitated by the protein kinase R (PKR) kinase and 2′-5′-oligoadenylate-dependent RNase L; II. RNA editing mediated by IFN-induced RNA-specific adenosine deaminase (ADAR1) and Mx protein GTPase; III. Induction of inducible nitric oxide synthase (iNOS2) and major histocompatibility complex class I and II proteins [62]. Furthermore, the activation of RLR signaling leads to IFN-dependent and non-IFN-dependent apoptosis of target cells, which is crucial for inhibiting viral replication [63,64]. MicroRNAs that target RIG-I/MDA5 can modulate RLR signaling and thereby impact viral replication. Therefore, in this section, we provide a comprehensive summary of the role of microRNAs in viral infections through their targeting of RIG-I/MDA5 to regulate RLR signaling.

While several proteins have been reported to modulate the RLR pathway and subsequent IFN signal transduction [65,66], studies on the regulation of this pathway by microRNAs remain limited. The RLR signaling pathway may give rise to certain microRNAs, such as miR-146a and miR-4661 [67,68], which in turn regulate the expression or function of various components of the RLR signaling pathway and IFN-I through a feedback mechanism. Additionally, RIG-I directly regulates the replication of vesicular stomatitis virus (VSV) in conjunction with miR-582-3p [69]. Furthermore, miR-92a has been found to diminish the VSV-induced production of IFN-I and promote viral replication in macrophages by targeting RIG-I [70]. Notably, downregulation of miR-340-5p has been reported in A549 cells infected with influenza A virus (IAV), and it reduces viral replication by targeting RIG-I and 2′-5′-oligoadenylate synthetase 2 (OAS2) [71].

Furthermore, miR-136 serves as a RIG-I immune agonist [72]. In general, miR-136 exhibits two functions in regulating host innate immunity against viruses. Another study reported that miR-4423-3p promoted the replication of the hepatitis C virus (HCV) by inhibiting the activation of the IFN pathway via RIG-I targeting [73]. However, whether miR-4423-3p blocks downstream molecules due to RIG-I inhibition only, or whether other mechanisms are also involved, remains to be investigated. Correspondingly, in response to viral infection, the host produces miR-485, which targets and destroys the RIG-I mRNA, decelerating the antiviral response and accelerating viral multiplication. Consequently, the host suppresses the expression of miR-485, which leads to markedly reduced Newcastle disease virus (NDV) and H5N1 influenza virus multiplication in mammalian cells. Unexpectedly, miR-485 also inhibits the replication of the H5N1 virus by binding to the RNA polymerase encoded by the H5N1 gene polymerase basic protein 1 (PB1) in a sequence-specific manner. In addition, miR-485 demonstrates dual selectivity: during elevated levels of the H5N1 virus, it targets PB1 in the cells, while during low levels of the virus, it targets RIG-I in the cells [74]. Therefore, miR-485 plays a dual role in preventing inappropriate antiviral signaling activation and reducing influenza virus infection.

Studies have reported that miR-146a holds potential as a novel biomarker for distinguishing between the acute and post-acute phases of COVID-19 [75]. However, a study reported that the hepatitis B virus (HBV) worsened the HBV infection by causing miR-146a to target RIG-I and RIG-G (retinoic acid-inducible gene G), thereby facilitating the negative regulation of the RIG-I signaling pathway [76]. Therefore, it is important to explore whether SARS-CoV-2 and other viruses are capable of immune escape via miR-146a.

Nevertheless, a fascinating investigation revealed that the Epstein–Barr virus (EBV)-encoded miR-BART6-3p, which targets RIG-I, promoted viral infection and was linked to the immune surveillance evasion of EBV [77]. Another correlational study identified miR-218 as a novel virus-induced miRNA that interfered with RIG-I expression and impaired the production of interferons to promote viral immune escape [78]. These studies suggest that virus-encoded microRNAs may interact with the host genes to influence the host’s ability to combat viruses, although the precise underlying mechanism remains unknown to date.

On the other hand, the overexpression of miR-34b-5p inhibited the MDA5 signaling pathway but not the RIG-I signaling pathway, thereby promoting the proliferation of the avian leukosis virus subgroup J (ALV-J)-infected DF-1 cells and facilitating the replication of ALV-J [79]. Recent findings indicate that microRNAs may play a role in regulating RLR signaling in the miiuy croaker. For instance, miR-203 was ascertained to target the 3′-UTR of MDA5, which then activated the downstream genes IRF3/7 and ultimately promoted the production of IFN-I and inflammatory cytokines and prevented the viral infection [80].

#### 2.1.2. Targeting Other Genes of RLR Signaling

Apart from targeting RIG-I/MDA5 to modulate RLR signaling during viral infections, microRNAs also exhibit similar effects by targeting other molecules within the RLR signaling pathway. For example, extensive research has highlighted the functional involvement of miR-33 in cellular metabolism, particularly in the maintenance of cholesterol homeostasis [81,82]. However, a study confirmed that overexpression of miR-33 impairs the RIG-I signaling pathway, strengthens the VSV viral load and lethality, and decreases the production of IFN-I in vitro and in vivo. This is achieved by targeting AMP-activated protein kinase (AMPK) [83]. Interestingly, cyprinid herpesvirus 3 (CyHV-3) infection reportedly induced another microRNA named miR-155, which targets AMPK. AMPK plays a crucial role in impeding the formation of activation aggregates by MAVS, consequently inhibiting the autophagy-mediated elimination of damaged mitochondria, and disrupting mitochondrial homeostasis. This disruption hinders the effective activation of MAVS, leading to a reduction in interferon expression. The research demonstrated that miR-155 exhibited anti-CyHV-3 activity via modulating the AMPK-MAVS-IFN axis and could, therefore, be utilized in studies focused on designing novel anti-CyHV-3 drugs [84].

Similar to miR-146a, let-7b, a microRNA, has been implicated in various conditions, including inflammation [85,86], cancers [87,88], and IgA nephropathy [89,90]. However, a recent study has revealed that let-7b can interfere with HCV infection by targeting the IκB kinase complex 1 (IKK1 or IKKα), leading to enhanced phosphorylation and nuclear translocation of IRF3 mediated by RIG-I. This ultimately results in the expression of IFN-β [91]. These findings suggest that let-7b plays a significant role in multiple disorders and could serve as a promising therapeutic target for the treatment of these diseases. In the context of IAV-infected A549 cells, the host protein contactin-1 (CNTN1) promotes viral replication by directly interacting with miR-200c, thereby inhibiting RIG-I-MAVS-mediated interferon signaling [92]. Rhabdovirus infection substantially upregulated the expression of host miR-3570 in Miichthys miiuy macrophages, and the induced miR-3570 promoted viral replication by targeting and downregulating MAVS [93]. Moreover, RIG-I/TBK1 signaling activation was induced in macrophages against HBV by the hepatic exosomes with decreased levels of miR-27b-3p. The expression level of miR-27-3p in serum exosomes shows promise as a potential biomarker for patients with chronic hepatitis B (CHB) [94]. Moreover, enterovirus infection leads to the downregulation of miR-526a, which targets cylindromatosis (CYLD) and suppresses the RIG-I-dependent innate immune response [95].

A separate investigation demonstrated that the upregulation of miRNA-146a in macrophages infected with VSV was dependent on both the RIG-I/NF-kB pathways. This miR-146a encourages VSV replication by targeting IRAK1, IRAK2, and TRAF6 and by preventing the release of IFN-I that is activated by the virus [67]. On the other hand, the hepatitis A virus (HAV) partly interferes with RIG-I/MDA5-mediated IFN-I signaling by targeting the vital adaptor molecule TRAF6 through miR-146a-5p [96]. The afore-stated study demonstrated that viruses might induce the production of miR-146a to evade host immune responses. Li et al. reported that miR-9-5p inhibits the replication of enterovirus 71 (EV71) by targeting the NF-κB-mediated innate immune response [97].

Notably, miR-30a inhibits TRIM25 expression and TRIM25-mediated RIG-I ubiquitination to enhance the replication of coxsackievirus B3 (CVB3) [98]. In addition, miR-202-5p was also reported to target the TRIM25-mediated ubiquitination of RIG-I and promote red spotted grouper nervous necrosis virus (RGNNV) infection in zebrafish [99]. Furthermore, TRIM25 is involved in the regulation of MDA5 expression. MiR-30a and miR-202-5p target TRIM25 and can activate both RIG-I and MDA5, thereby augmenting the antiviral response.

The miR-302/367 cluster is one of the most studied microRNAs and consists of five members: miR-367, miR-302a, miR-302b, miR-302c, and miR-302d. The miR-302/367 cluster is widely involved in cellular differentiation and development, tumor progression, immune regulation, and other biological processes [100,101]. MiR-302b, as a member of this family, has been shown to control MAVS-mediated antiviral innate immunity by targeting mutations in the solute carrier family 25, member 12 (SLC25A12) transporter protein to regulate the mitochondrial metabolism [102]. MiR-302c mediates IAV-induced IFN-β expression by targeting NF-κB [103]. In addition, miR-302a, another member of the miR-302/367 cluster, inhibits IRF5 expression by directly targeting the IRF5 3′-UTR, leading to the overexpression of inflammatory cytokines and chemokines [104]. Thus, the miR-302/367 cluster could serve as a potential regulator of virus-induced cytokine storms and provide a candidate target for the treatment of viral infections.

The above-stated findings provide fresh insights into how microRNAs are involved in the regulation of RLR signaling. Therefore, this field of research is of great importance in the development of novel antiviral therapies in the future.

#### 2.1.3. The Role of microRNAs and RLRs in COVID-19 Sensing

The infection of COVID-19, caused by the severe acute respiratory syndrome coronavirus 2 (SARS-CoV-2), involves the activation of the host immune response through the recognition of the viral RNA by RIG-I/MDA5. Consequently, our research focuses on investigating the involvement of RLR signaling and microRNAs in the infection process. The two cytoplasmic viral RNA sensors, RIG-I and MDA5, are vital for the immune escape of SARS-CoV-2. Several coronavirus-encoded proteins have been confirmed to inhibit the IFN response, although the precise mechanism remains to be elucidated [105,106]. Nonetheless, the SARS-CoV-2 membrane (M) protein might bind RIG-I/MDA5 and limit the production of IFN-I and IFN-III, which reveals a role for this protein in antagonizing IFN-mediated innate viral immunity and viral replication [107]. The major protease (Mpro) of SARS-CoV-2, known as NSP5, has garnered attention as a potential therapeutic target for COVID-19 treatment. NSP5 exhibits unique cleavage specificity, as it not only cleaves lengthy viral polypeptides but also targets nucleotide-binding oligomerization-like receptor protein 12 (NLRP12) and TAB1. This distinct cleavage profile differentiates NSP5 from known human proteases [108]. During SARS-CoV-2 infection, the nucleoprotein (N protein) is the most abundant viral protein. It consists of an N-terminal structural domain (NTD) and a C-terminal structural domain (CTD). These domains are primarily responsible for RNA binding and are crucial for assembling the viral genome into viral particles [109]. Attractively, the SARS-CoV-2 NSP5 and N protein reduce the formation of antiviral stress granules (avSGs). NSP5 also disrupts RIG-I signaling by destroying the RIG-I-MAVS complex, thereby reducing the antiviral response [110]. The above results indicate that RLR signaling (particularly RIG-I and MDA5) is the main pathway through which SARS-CoV-2 escapes the innate immune response of the host.

Previous studies have extensively documented the importance of microRNAs in viral infections. Two key aspects have been elucidated: Firstly, host microRNAs enhance the immune response, thereby impacting viral replication. Secondly, viral microRNAs have been demonstrated to target host immune cells, impeding viral clearance [111,112]. Therefore, the identification of microRNAs that are strongly correlated with SARS-CoV-2 may provide fresh insights into the diagnosis and prognosis of COVID-19, and could be used as a feasible strategy to develop novel targeted therapies. The upregulation of miR-146a and miR-155 in the oral cavities of patients with diabetes and periodontitis, as reported by Jelena R. Roganovic, has been suggested to potentially enhance the expression of angiotensin-converting enzyme 2 (ACE2) and alter the host’s antiviral response to the SARS-CoV-2 virus [113]. Nevertheless, further verification is required to confirm this hypothesis.

In a recent study, a correlation was observed between the upregulation of hsa_circ_0000479, RIG-I, and IL-6 and the downregulation of has-miR-149-5p, suggesting that the has-circ-0000479/hsa-miR-149-5p/RIG-I/IL-6 axis may be involved in regulating the immune response to SARS-CoV-2 [114]. As a whole, the findings imply that overexpression of hsa_circ_0000479 may affect the expression of IL-6 and RIG-I via sponging hsa-miR-149-5p in COVID-19. Sponging refers to the phenomenon where circular RNAs (circRNAs), which possess multiple miRNA binding sites, act as competitive endogenous RNAs (ceRNAs) by specifically sequestering miRNAs. This process can be likened to a sponge absorbing water, as circRNAs “isolate” miRNAs from their target mRNAs, thus impacting the negative regulation of target mRNAs by miRNAs [115,116,117]. The regulation of the expression of IL-6 and RIG-I, which are two genes essential for the immunological response to COVID-19 infection, could impact the severity of COVID-19 symptoms (Figure 3). Therefore, the hsa-circ-0000479/hsa-miR-149-5p/IL-6 RIG-I axis could serve as a promising therapeutic target in the treatment of COVID-19.

Furthermore, several other microRNAs have been implicated in relation to COVID-19, including miR-1207-5p [118], miR-320 [119], miR-200c-3p [120], miR-146a [121], miR-421-5p [122], and let-7b [123], among others. Importantly, microRNAs encoded by SARS-CoV-2 have the potential to target host cell mRNAs involved in viral replication, thereby facilitating viral replication and regulating host gene expression [124,125,126]. The elucidation of the detailed mechanism underlying this effect may facilitate the development of microRNA attenuated vaccines against COVID-19.

### 2.2. Regulation of RLR Signaling by MicroRNAs in Cancer

MicroRNAs in RLR signaling are also involved in regulating cancer development. Once activated, the RIG-I pathway will induce target cell death through a variety of pathways, including endogenous apoptosis, exogenous apoptosis, pyroptosis, and autophagy [127,128,129]. MicroRNAs have been established to modulate tumor progression through the regulation of RLR signaling. Notably, microRNA-545 has been demonstrated to exert significant effects in various cancers, including breast cancer [130], bladder cancer [131], ovarian cancer [132], cervical cancer [133], and other malignancies [134,135,136]. However, further investigation demonstrated that the overexpression or knockdown of miR-545 could promote or inhibit cancer cell proliferation by targeting RIG-I [137,138,139]. Moreover, the inhibition of PTEN (phosphatase and tensin homolog), a tumor suppressor gene frequently mutated, deleted, and functionally inactivated in human cancers, along with RIG-I, has been shown to enhance the phosphatidylinositol 3-kinase (PI3K)-protein kinase B (AKT) signaling pathway. This regulation involves the intronic microRNAs miR-374b and miR-545, which are modulated by the intronic miR-421 [140]. Therefore, it is inferred that the various loci that regulate the functional components of tumor progression are relatable. However, little research has been conducted in the area, and the underlying mechanisms also remain unknown so far; therefore, it is necessary to explore novel investigations in this aspect of cancer treatment and prognosis research.

### 2.3. Others

Except for the above two aspects, studies have also reported that certain microRNAs stimulate the RIG-I signaling pathway to affect IFN-I production. For instance, miR-139 promotes IFN secretion in prostate cancer cells by acting as an immunological agonist of RIG-I [141]. Moreover, miR-1248 was reported to activate IFN-β through a direct association with RIG-I and AGO2 (argonaute 2) [142]. Remarkably, miR-145 was transfected into human mesenchymal stem cells (hBM-MSCs) and human articular chondrocytes (hSCs), where it could induce immune responses by targeting RIG-I [143].

Similarly, studies have revealed that miR-145-5p, miR-122, and miR-489 can target MDA5, DAK, and TRAF6, respectively, in the miiuy croaker to modulate the RLR signaling pathway upon poly(I:C) stimulation [144,145,146]. These findings offer valuable insights into the regulatory role of microRNAs in the immune response of fish. A comprehensive summary of the microRNAs involved in the regulation of the RIG-I signaling pathway, as discussed in the preceding sections, can be found in Figure 4 and Table 1.

## 3. Conclusions and Prospects

The present review summarizes several roles that microRNAs play in the regulation of various physiological processes, such as cell division, apoptosis, etc. During recent years, numerous microRNAs have been discovered through advancements in high-throughput sequencing technology. Moreover, the regulatory functions of microRNAs have been receiving greater attention all over the world. Innate antiviral immune responses are mainly mediated by RLRs, particularly RIG-I/MDA5. Modulating RLR signaling through noncoding RNAs presents a promising approach for optimizing the immune response against viral infections and inhibiting tumor progression. The regulatory role of miRNAs in RLR signaling primarily involves binding to the 3′ UTR of the mRNAs associated with RLR signaling, thereby modulating the mRNA levels of target genes and subsequently influencing protein expression. This, in turn, enhances or suppresses RLR signaling, leading to the secretion of IFNs and inflammatory cytokines that impact viral replication and cancer progression. While previous studies have demonstrated the regulatory impact of microRNAs on RLR signaling, further investigation is required to elucidate the intricacies of their interactions.

The “energy factories” within the cell, mitochondria are essential organelles in the metabolic processes of the cell. Studies have demonstrated that the mitochondrial route may be used by microRNAs to regulate cellular processes. For instance, in adipose tissue, miR-494-3p directly reduces PGC1 expression and the ensuing mitochondrial biogenesis [147]. MiR-125 promotes the demise of cancer cells via regulating mitochondrial dynamics and metabolism [148]. Furthermore, overexpression of miR-25 dramatically decreased the levels of mitochondrial calcium uniporter (MCU) expression, hindered mitochondrial Ca^2+^ uptake, and prevented apoptosis in cancer cells [149]. The control of mitochondrial calcium homeostasis, mitochondrial metabolism, and mitochondrial biogenesis are all critical processes that are impacted by microRNAs. Targeting the mitochondrial pathway may be an effective potential strategy for disease treatment.

RIG-I/MDA5 induces antitumor immune responses through the IFN-dependent activation of effector T cells. On the other hand, RIG-I/MDA5 also induces cancer cell apoptosis in an IFN-independent manner. Therefore, targeting RIG-I/MDA5 has several advantages over conventional cancer therapies. First, RIG-I/MDA5 may be activated using the corresponding ligands and, therefore, significant tumor-cell-specific apoptosis may be induced. Second, the expression of RIG-I/MDA5 is also associated with the prognosis of certain cancers. For instance, liver cancer patients with low RIG-I expression have a poor prognosis [150], while ovarian cancer patients with high RIG-I expression have a similar outcome [151]. Although several RIG-I/MDA5 ligands have been employed in clinical trials, challenges are encountered when attempting to target RIG-I/MDA5 for cancer treatment. For instance, one challenge is to further optimize the methods to be used for enabling these ligands to reach the site of the cancer. Another challenge is the difficulty of controlling the dosage of the ligand. While a high dosage of the ligand that activates RIG-I/MDA5 may kill the cancer cells, low doses of the ligand may result in low levels of interferons and pro-inflammatory cytokines, which could accelerate cancer growth. Moreover, the activation of RIG-I/MDA5 could activate the NF-κB signaling pathway, thereby aggravating inflammation. Activation of RIG-I/MDA5 through endogenous pathways, such as microRNAs and long noncoding RNAs (lncRNAs), might enable achieving better effects and could, therefore, serve as a novel strategy for cancer treatment. Unfortunately, studies so far have not reported any microRNAs that regulate LGP2, another member of the RLR family.

The emergence of drug resistance to anticancer medications is a major obstacle to successful treatment. MicroRNAs have gained attention due to their ability to regulate genes involved in the cellular responses to drugs. Several miRNAs have been identified to sensitize cancer cells to therapy and serve as biomarkers of drug resistance. For example, a study investigating the co-delivery of miR-345 and gemcitabine in pancreatic cancer demonstrated superior outcomes compared with treatment with either miR-345 or gemcitabine alone [152]. Similar positive results were observed when combining doxorubicin with miR-34a for breast cancer treatment [153]. Therefore, the combination of RLR agonists and microRNAs in tumor immunotherapy may significantly enhance treatment effectiveness, although further investigation is warranted.

Moreover, microRNAs and RLRs also have distinct roles in SARS-CoV-2 infection. While microRNAs such as miR-146a, let-7b, and miR-200c have been implicated in mediating RLR signaling during SARS-CoV-2 infection, it remains to be empirically validated whether these alterations in RLR signaling impact the infection itself.

MicroRNA-based therapies are promising. However, how to deliver microRNAs safely and effectively to target organs and target cells is the biggest challenge at present. Currently, the following major delivery modes have been developed, which may help miRNA therapeutics to move toward clinical application: (I) conjugation [154]; (II) virus-associated delivery [155]; (III) nanoparticles [156]; and (IV) exosome-associated delivery [157]. These delivery techniques are now being employed in therapies that have received clinical approval or are undergoing clinical studies. Due to their low cytotoxicity, low antigenicity, and capacity to avoid phagocytosis and the endocytosis route, exosomes may be the best delivery method of these [158,159]. However, dosage continues to be a key restriction in the area of miRNA therapy. Clinical trials have seldom been able to pinpoint the precise dose at which RNA treatment is successfully delivered to the cell type of interest.

In conclusion, the immune system utilizes various miRNAs to finely regulate its functional capacity, maintaining a delicate balance between activation and inhibition. However, the precise regulatory mechanisms of miRNAs in the RIG-I response are not fully understood, particularly how these miRNA networks collaboratively optimize the immune response. A deeper understanding of the miRNA-mediated regulation of RIG-I signaling will uncover novel therapeutic targets for inflammatory diseases and cancer.

## Figures and Tables

**Figure 1 biomolecules-13-01344-f001:**
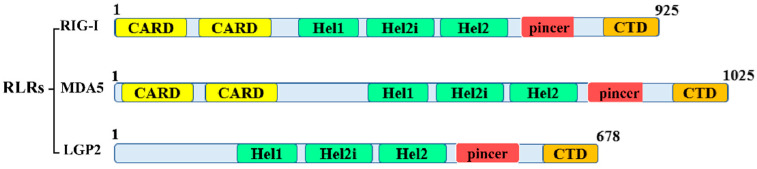
RIG-I-like receptor protein domain structure. The three RLRs are DExD/H-box helicase proteins of type SF2 that contain the conserved domains Hel1 and Hel2 in their core DExD/H-box helicase domain. The Hel2i insertion domain, essential for RIG-I autoregulation, is located between the two helicase domains. A broadly conserved C-terminal region is also present in all RLRs (CTD). A positively charged binding pocket is present only in the CTD of RIG-I to identify 5′-PPP or 5′PP RNA substrates. Two N-terminal caspase activation and recruitment domains are shared by RIG-I and MDA5 (CARDs). For interactions with MAVS and the start of downstream antiviral signaling, several CARDs must be aligned.

**Figure 2 biomolecules-13-01344-f002:**
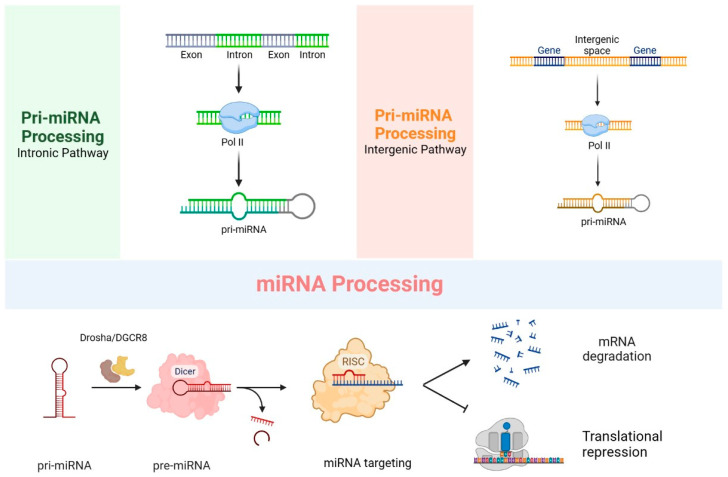
Biological processes of miRNAs. Pri-miRNAs can be formed via the intronic pathway and the intergenic pathway. Processing of pri-miRNAs produces pre-miRNAs. Pre-miRNAs are cleaved by the enzyme Dicer and then re-digested to generate mature miRNAs, which can target mRNAs, resulting in mRNA degradation or translational repression.

**Figure 3 biomolecules-13-01344-f003:**
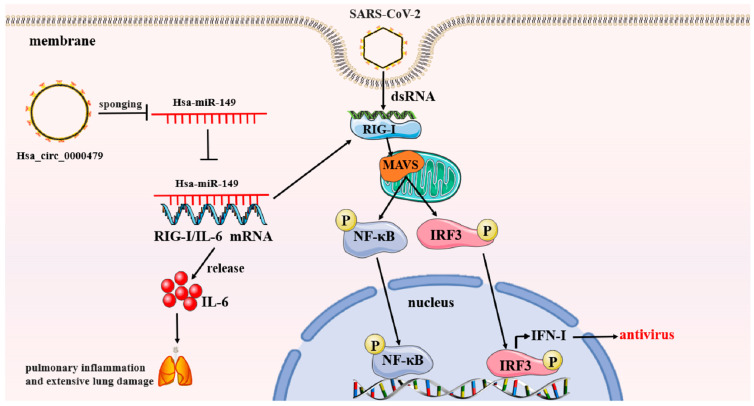
The hsa_circ_0000479/hsa-miR-149/RIG-I, IL-6 axis regulates SARS-CoV-2 infection. Up-expressed hsa_circ_0000479 could modulate the expression of IL-6 and RIG-I through sponging of hsa-miR-149–5p. On the one hand, activation of the RIG-I signaling pathway promotes the release of IFN-I, which exerts antiviral effects. On the other hand, the expression level of IL-6 is associated with lung inflammation and extensive lung injury.

**Figure 4 biomolecules-13-01344-f004:**
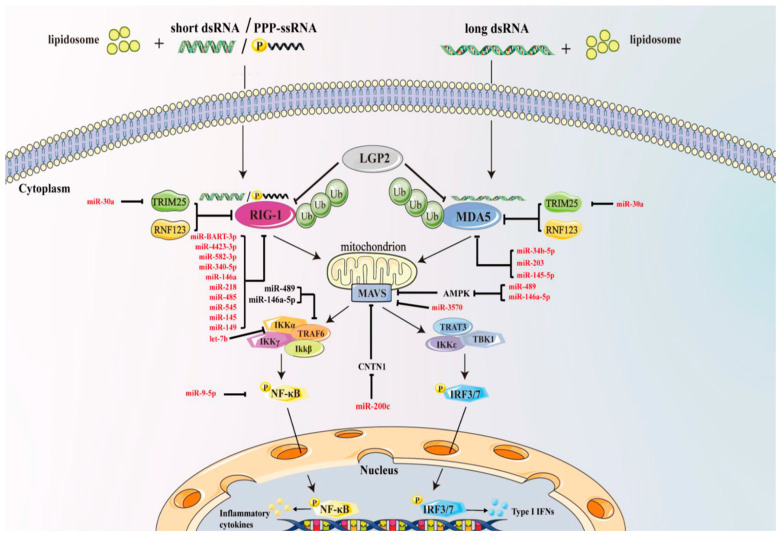
MicroRNAs regulate the RIG-I-like receptor signaling pathway. RIG-I and MDA5 recognize a complementary set of viral cytosolic dsRNA ligands. Their activation is tightly regulated by phosphorylation, ubiquitination, and host proteins such as LGP2. RIG-I and MDA5 signal to MAVS, and then induce NF-κB and IRF3/7 to enter the nucleus, which promotes the release of type I IFNs and inflammatory cytokines. The activity of RIG-I and MDA5 can be regulated by several E3 ubiquitin ligases, including TRIM25 and RNF123. The circled ‘P’ indicates phosphorylation, and the circled ‘Ub’ indicates ubiquitination. MiR-BART6–3p, miR-4423–3p, miR-340–5p, miR-545, miR-145, miR-146a, miR-218, miR-485, miR-149, miR-92a, and miR-582–3p target RIG-I; miR-34b-5p, miR-203, and miR-145-5p target MDA5; miR-122 targets DAK to inhibit the expression of MDA5; miR-30a and miR-202-5p target TRIM25; miR-155 and miR-33 target AMPK; miR-200c targets CNTN1 to regulate MAVS, while miR-3570 directly targets MAVS; miR-489 and miR-146a-5p target TRAF6; and miR-9–5p targets NF-κB. MicroRNA let-7b targets IKKα. These microRNAs directly or indirectly regulate the RIG-I-like receptor signaling pathway to play antiviral, antitumor, or immunomodulatory roles.

**Table 1 biomolecules-13-01344-t001:** MicroRNAs involved in the regulation of RLR signaling pathway, their targets, and results mentioned in this paper.

MicroRNAs	Targets	Results	Reference
miR-92a	RIG-I	Lowering VSV-induced production of type I IFN and promoting viral replication in macrophages	[70]
miR-218	RIG-I	Impairing interferon production to promote viral immune escape	[78]
miR-4423-3p	RIG-I	Promoting HCV replication	[73]
miR-BART6-3p	RIG-I	Promoting EBV infection Limiting influenza virus infection and avoiding erroneous antiviral signaling activation	[77]
miR-485	RIG-I	Inhibiting pancreatic ductal adenocarcinoma growth	[76]
miR-545	RIG-I	Promoting the proliferation of hepatocellular carcinoma cells; inhibiting the proliferation and migration of oral squamous cell carcinoma	[137,138,139]
miR-145	RIG-I	Inducing immune responses by targeting RIG-I expression	[143]
miR-582-3p	RIG-I	Controlling VSV replication	[69]
miR-340-5p	RIG-I, OAS2	Inhibiting IAV replication	[71]
miR-146a	RIG-I, RIG-G	Aggravating HBV infection	[76]
miR-34b-5p	MDA5	Promoting the proliferation and migration of DF-1 cells and ALV-J replication	[79]
miR-203	MDA5	Negatively regulating RLR signaling pathway	[80]
miR-145-5p	MDA5	Regulating RLR signaling pathway	[144]
miR-122	DAK	Inhibiting the expression of MDA5	[145]
miR-30a	TRIM25	Regulating type I IFN response and promoting CVB3 replication	[98]
miR-202-5p	TRIM25	Inhibiting RIG-I-dependent innate immune responses to RGNNV infection	[99]
miR-3570	MAVS	Inhibiting the expression of MAVS and promoting the replication of rhabdovirus	[93]
miR-200c	CNTN1	Inhibiting the expression of MAVS and promoting the replication of IAV	[92]
miR-155	AMPK	Inhibiting cyprinid herpesvirus 3 replication via regulating the AMPK-MAVS-IFN axis	[84]
miR-33	AMPK	Inhibiting the activation of MAVS through AMPK	[83]
miR-489	TRAF6	Negatively regulating TRAF6 and involved in the immune response to poly(I:C) stimulation	[146]
miR-146a-5p	TRAF6, IRAK1, IRAK2	Negatively regulating VSV-triggered type I IFN production, thus promoting VSV replication in macrophages,	[96]
let-7b	IKKa	leading to an increase in IFN-I expression	[91]
miR-9-5p	NF-κB	Restraining EV71 replication	[97]

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
