# Peer review of "MicroRNAs in the Regulation of RIG-I-like Receptor Signaling Pathway: Possible Strategy for Viral Infection and Cancer"

_biomolecules, 2023, doi:10.3390/biom13091344_

Round 1
Reviewer 1 Report
Mitochondria, cellular powerhouses of eukaryotes, are known to act as central hubs for multiple signal transductions and recent research reveals that the organelles are involved in cellular innate antiviral immunity in mammals. Mitochondrial-mediated antiviral immunity depends on the activation of RIG-I signal transduction pathway and on the participation of a mitochondrial outer membrane adaptor protein, MAVS. In the manuscript, the authors have reviewed an interesting topic on functional relationship between microRNAs and RIG-I signaling. Overall, the manuscript is a well written comprehensive review on the topic. I recommend the manuscript is well suitable for publication in the journal, and a couple of minor issues/comments will further strength their overview.
1). Molecular mechanisms of miRNAs in its actions, especially involving in mitochondrial metabolism, are mostly lacking. I sure several groups studying on miRNAs involves in mitochondrial functions. The authors should discuss how the miRNAs control cellular function via mitochondrial pathway.
2). I would suggest the authors should mention the role of miR-302 family in the revised manuscript. So far, three studies are described in depth (PMC2573233; PMC5766960; PMC6956542).
3). A major challenge in applying the miRNA-based therapies is delivering these compounds to the infected site in controlled amounts without inducing toxicity. I would suggest the authors should also mention and discuss how to deliver useful candidates into our body.
N/A
Author Response
Dear Reviewer 1:
Thank you for your comments concerning our manuscript entitled " MicroRNAs in the regulation of RIG-I-like receptor signaling pathway: possible strategy for viral infection and cancer "(Manuscript ID: biomolecules-2562947). Those comments are all valuable and very helpful for revising and improving our paper, as well as the important guiding significance to our research. We have studied the comments carefully and have made corrections which we hope meet with approval. Revised portions are marked in red in the paper. The main corrections in the paper and the responses to the reviewer's comments are as following:
Mitochondria, cellular powerhouses of eukaryotes, are known to act as central hubs for multiple signal transductions and recent research reveals that the organelles are involved in cellular innate antiviral immunity in mammals. Mitochondrial-mediated antiviral immunity depends on the activation of RIG-I signal transduction pathway and on the participation of a mitochondrial outer membrane adaptor protein, MAVS. In the manuscript, the authors have reviewed an interesting topic on functional relationship between microRNAs and RIG-I signaling. Overall, the manuscript is a well written comprehensive review on the topic. I recommend the manuscript is well suitable for publication in the journal, and a couple of minor issues/comments will further strength their overview.
1). Molecular mechanisms of miRNAs in its actions, especially involving in mitochondrial metabolism, are mostly lacking. I sure several groups studying on miRNAs involves in mitochondrial functions. The authors should discuss how the miRNAs control cellular function via mitochondrial pathway.
Response: Thank you for your suggestion, we have added to our discussion about the control of cellular functions by miRNAs through the mitochondrial pathway, including the regulation of mitochondrial biosynthesis, energy metabolism and calcium homeostasis, suggesting that targeting the mitochondrial pathway may be a potential strategy for disease treatment. (lines 10-20 of page 11)
2). I would suggest the authors should mention the role of miR-302 family in the revised manuscript. So far, three studies are described in depth (PMC2573233; PMC5766960; PMC6956542).
Response: Thank you for your valuable comments. We have mentioned the role of miR-203 family in the revised manuscript. (lines 51-55 of page 6 to lines 1-7 of page 7)
3). A major challenge in applying the miRNA-based therapies is delivering these compounds to the infected site in controlled amounts without inducing toxicity. I would suggest the authors should also mention and discuss how to deliver useful candidates into our body.
Response: Thank you for your valuable comments. This is crucial for microRNA-based therapies, so we have mentioned in our discussion the 4 main systems currently used for microRNA delivery, which may help microRNAs to be used in the clinic. (lines 1-11 of page 12)
Reviewer 2 Report
I have gone through the article and have some minor suggestions for authors:
1. Authors should explain every abbreviation used for the first time at place.
2. Kindly check for TRAF-6 explanation.
3. On page 3 lines 19-36, kindly cite appropriate abbreviations as the current reference number 33 is seems wrong to me.
4. On page 3, line 32-33, kindly cite a reference there.
5. On page 6, lines 2-3 also need a reference as per the information.
6. page 6 lines 14-16 also need an appropriate reference.
On page 7, line 20-21 need references.
Also, on page 3, in lines 35 and 36, authors should provide information on other regulators of RLR signaling and then correlate the introduction of miRNAs as RLR signaling regulators. Currently, it is a direct jump miRs.
I will recommend authors to have check on English language for minor errors.
Author Response
Dear Reviewer 2:
Thank you for your comments concerning our manuscript entitled " MicroRNAs in the regulation of RIG-I-like receptor signaling pathway: possible strategy for viral infection and cancer "(Manuscript ID: biomolecules-2562947). Those comments are all valuable and very helpful for revising and improving our paper, as well as the important guiding significance to our research. We have studied the comments carefully and have made corrections which we hope meet with approval. Revised portions are marked in red in the paper. The main corrections in the paper and the responses to the reviewer's comments are as following:
I have gone through the article and have some minor suggestions for authors:
- Authors should explain every abbreviation used for the first time at place.
Response: Thanks for the excellent advice! We've explained each of the abbreviations when they first appear. For example, melanoma differentiation-associated gene 5 (MDA5), and laboratory of genetics and physiology 2 (LGP2), etc.
- Kindly check for TRAF-6 explanation.
Response: We apologize for our error, so we have corrected it to “tumor necrosis factor receptor associated factor 6 (TRAF6)”. (line 3 of page 3)
- On page 3 lines 19-36, kindly cite appropriate abbreviations as the current reference number 33 is seems wrong to me.
Response: Thank you for your suggestion! We apologize for the disorganized reference alignment due to our negligence, we have carefully checked the references and corrected them to ensure that they are now cited correctly.
- On page 3, line 32-33, kindly cite a reference there.
Response: Thank you for your suggestion! We have cited the references in the appropriate places. (line 37 of page 3)
- On page 6, lines 2-3 also need a reference as per the information.
Response: Thank you for your suggestion! We have cited the references in the appropriate places. (line 5 of page 6)
- page 6 lines 14-16 also need an appropriate reference.
Response: We apologize that the original reference was misplaced due to our negligence, we have corrected it and the corresponding reference is now correct.
On page 7, line 20-21 need references.
Response: We apologize that our presentation has caused you to misunderstand. This sentence here is a summary of the previous reference “[99-102]”. Therefore, we have changed “These” to “The above”. (line 32 of page 7)
Also, on page 3, in lines 35 and 36, authors should provide information on other regulators of RLR signaling and then correlate the introduction of miRNAs as RLR signaling regulators. Currently, it is a direct jump miRs.
Response: what a great suggestion! For the regulation of RLR signaling, LGP2 and E3 ubiquitin ligases are crucial, however, more and more studies have also confirmed the involvement of microRNAs in the regulation of RLR signaling. Therefore, we have added a sentence “In addition to the regulation of RLR signaling through LGP2 and E3 ubiquitin ligases, increasing evidence confirms that microRNAs are also involved in the regulation of RLR signaling” here to make the connection between the previous and the next content more coherent. (lines 40-42 of page 3)
In addition, we have checked our English language.
Reviewer 3 Report
I find this paper very interesting and a complex summary of the importance of RIG-like receptors and miRNA relation. The paper is written clearly and the figures and tables are well constructed. I have no issues about this paper, I recommend it to be published.
Author Response
Dear reviewer 3,
Thank you for your recognition of our manuscript, and we hope that our manuscript will provide some valuable references for the field.
Round 2
Reviewer 1 Report
This MS is ready for publication.
Author Response
Thank you for your review.